# Nobiletin Attenuates Adipogenesis and Promotes Browning in 3T3-L1 Adipocytes Through Exosomal miRNA-Mediated AMPK Activation

**DOI:** 10.3390/cimb48010036

**Published:** 2025-12-26

**Authors:** Shweta Chauhan, Hana Baek, Varun Jaiswal, Miey Park, Hae-Jeung Lee

**Affiliations:** 1Institute for Aging and Clinical Nutrition Research, Gachon University, Seongnam 13120, Republic of Korea; chauhanshweta210@yahoo.com (S.C.); computationalvarun@gmail.com (V.J.); 2Department of Food and Nutrition, College of BioNano Technology, Gachon University, Seongnam 13120, Republic of Korea; 3Department of Food Science and Biotechnology, Gachon University, Seongnam 13120, Republic of Korea; ruru456123@gachon.ac.kr; 4Department of Health Sciences and Technology, Gachon Advanced Institute for Health Sciences & Technology (GAIHST), Gachon University, Incheon 21999, Republic of Korea

**Keywords:** phytochemical, nobiletin, exosomal miRNAs, anti-obesity activities

## Abstract

Nobiletin, a citrus-derived polymethoxylated flavone, has been reported to exert anti-obesity effects, but its molecular mechanisms remain poorly understood. This study aimed to investigate whether nobiletin suppresses adipogenesis and promotes browning in 3T3-L1 adipocytes by modulating exosomal microRNAs (miRNAs) and AMPK signaling. To this end, we treated 3T3-L1 adipocytes with various concentrations of nobiletin and evaluated gene and protein expression by RT-qPCR and Western blotting. Nobiletin significantly reduced intracellular lipid accumulation at 50 μM (*p* < 0.001) and downregulated key adipogenic transcription factors, PPARγ, C/EBPα, and SREBP-1c, and suppressed the lipogenic enzyme FAS, while activating the AMPK/ACC signaling pathway. Concomitantly, it enhanced the expression of thermogenic markers UCP-1, PRDM16, and PGC-1α, indicating a metabolic shift toward energy expenditure. Exosomal RNA-seq revealed 10 differentially expressed miRNAs, of which miR-181d-5p (3.1-fold) and miR-221-3p (2.4-fold) were upregulated, whereas miR-205-5p (−2.9-fold), miR-331-3p (−3.2-fold), miR-130b-3p (−2.6-fold), miR-143-5p (−2.9-fold), miR-183-3p (−2.8-fold), miR-196b-5p (−2.4-fold), miR-26b-3p (−2.2-fold), and miR-378d (−2.7-fold) were verified by RT-qPCR after nobiletin treatment (50 μM). These miRNAs are functionally associated with adipogenic and thermogenic pathways, supporting a regulatory role of the exosomal miRNA network in nobiletin’s action. Collectively, our results identify a novel exosome–miRNA–AMPK axis underlying the anti-adipogenic and browning-inducing activities of nobiletin, highlighting its potential as a therapeutic phytochemical for obesity prevention.

## 1. Introduction

Obesity is a major global health burden, responsible for an estimated 2.8 million deaths annually. More than 650 million adults are classified as obese, and prevalence continues to rise in children and adolescents as well [1]. Obesity significantly increases the risk of severe comorbidities, including cardiovascular disease, diabetes, and various cancers, and exacerbates complications across numerous pathological conditions [2,3,4]. It is a complex metabolic disease linked to several genes, biological pathways, and other factors, such as diet and lifestyle [5]. The complex association between different factors and the pathology of obesity makes treatment challenging. The approved synthetic drugs against obesity, such as orlistat, have been associated with gastrointestinal adverse effects, suggesting the possibility of developing natural compounds as relatively safe drug candidates against obesity, as long-term medication is required [6]. Medicinal plant phytochemicals are essential for treating various diseases and health conditions, including obesity [7,8,9,10,11,12,13,14].

Nobiletin, a polymethoxylated flavone, is a critical phytocompound for citrus plants with important biological properties, including anti-obesity effects in cell lines and animal models [15,16]. In complex disorders such as obesity, knowledge of the mechanism of action is required to effectively develop drug candidates as efficient therapeutics [17]. There is rising apprehension about the role of miRNAs in complex diseases such as cancer and metabolic diseases, including obesity, and the possibility of targeting them [18,19]. The role of nobiletin in various cancers has been widely studied, particularly regarding its expression and modulation by miRNAs [20,21,22]. Nobiletin has been shown to exert anti-cancer activity through miRNA regulation, including anti-breast cancer effects [21,23], inhibition of immune evasion in lung cancer [24], and restoration of dysregulated miRNA expression in skin cancer [20].

Nobiletin has demonstrated a wide range of anti-obesity and metabolic regulatory effects in various experimental models. In high-fat diet-induced obese mice, nobiletin reduced body weight gain, improved insulin sensitivity, and enhanced glucose tolerance [25,26]. It also prevented hepatic lipid accumulation and steatosis by suppressing lipogenic gene expression and activating AMPK in the liver and adipose tissue [27]. Additionally, nobiletin modulates gut microbiota composition and reduces chronic inflammation associated with obesity-related metabolic dysfunction [15]. These findings collectively highlight nobiletin’s multifaceted actions against obesity and related disorders, such as type 2 diabetes and non-alcoholic fatty liver disease (NAFLD). However, the mechanisms underlying its exosomal miRNA-related functions remain poorly understood.

The involvement of miRNAs in obesity is well established, given their roles as key post-transcriptional regulators and therapeutic targets. Recent evidence suggests that exosomes can enter adipocytes and modulate adipogenesis by influencing critical signaling pathways that suppress lipid accumulation, highlighting their potential as an anti-obesity strategy [28]. To date, no studies have reported the anti-obesity effects of nobiletin mediated by miRNA regulation. Moreover, previous findings on nobiletin’s influence on adipogenesis in 3T3-L1 cells have been inconsistent. Therefore, this study aimed to evaluate the anti-obesity activity of nobiletin in 3T3-L1 adipocytes and to characterize changes in obesity-related genes and proteins. Notably, we also performed a comprehensive analysis of exosomal miRNAs in both nobiletin-treated and control groups.

This study revealed that nobiletin modulates key genes and signaling pathways involved in adipogenesis and lipid metabolism. Comparative miRNA profiling demonstrated that several exosomal miRNAs were differentially expressed following nobiletin treatment, establishing a mechanistic link between nobiletin and miRNA-mediated anti-obesity effects. This is the first study to demonstrate nobiletin’s anti-obesity effect via exosomal miRNA modulation in adipocytes. The functional roles of these miRNAs in obesity further support their potential as therapeutic targets for nobiletin-based intervention.

## 2. Materials and Methods

### 2.1. Chemicals and Materials

Nobiletin 5-Hydroxy-3,6,7,8,3′,4′-hexamethoxyflavone was purchased from Sigma (St. Louis, MO, USA) and dissolved in dimethyl sulfoxide (DMSO; Sigma-Aldrich, St. Louis, MO, USA). Bovine calf serum (BCS), Dulbecco’s Modified Eagle medium (DMEM), and trypsin-EDTA were purchased from Thermo Fisher Scientific (San Jose, CA, USA), and insulin was purchased from Thermo Fisher Scientific (San Jose, CA, USA). Dexamethasone (DEX; Sigma-Aldrich, St. Louis, MO, USA), 3-Isobutyl-1-methylxanthine (IBMX; Sigma-Aldrich, St. Louis, MO, USA), and Oil Red O (Sigma-Aldrich, St. Louis, MO, USA) were used. Cell viability was assessed using the Cell Counting Kit-8 (CCK-8; Dojindo Molecular Technologies, Rockville, MD, USA). Monoclonal antibodies against CCAAT/enhancer binding proteins α (C/EBPα), sterol regulatory element binding protein 1 (SREBP1), Peroxisome Proliferator-Activated Receptor γ (PPARγ), fatty acid synthase (FAS), uncoupling protein 1 (UCP-1), PR domain containing 16 (PRDM16), peroxisome proliferator-activated receptor gamma coactivator 1-α (PGC-1α), β-actin, acetyl-CoA carboxylase (ACC), p-ACC, AMP-activated protein kinase (AMPK), p-AMPK (1:1000, Cell Signaling Technology, Danvers, MA, USA), and horseradish peroxidase-coupled anti-rabbit or anti-mouse secondary antibodies were procured from Abcam (Cambridge, MA, USA). The protein measurement solution (PRO-MEASURE), Western blot detection system, and protein lysis buffer were obtained from iNtRON Biotechnology (Seongnam-si, Gyeonggi-do, Korea). Polyvinylidene Difluoride (PVDF) membranes were purchased from Merck (Burlington, MA, USA).

### 2.2. Cell Viability Assay

3T3-L1 preadipocytes were seeded at 1 × 10^4^ cells/well in 96-well plates and cultured for 24 h. Subsequently, the cells were treated with nobiletin and incubated for an additional 24 h at 37 °C in a 5% CO_2_ atmosphere. Cell viability was assessed using the Cell Counting Kit-8 assay (Dojindo Molecular Technologies, MD, USA) according to the manufacturer’s protocol. Absorbance was measured at 450 nm with a microplate reader (BioRad, Hercules, CA, USA). Results were expressed as a percentage of cell viability, and the experiment was performed in triplicate.

### 2.3. Cell Culture and Differentiation of Preadipocytes

The 3T3-L1 cells were obtained from the American Type Culture Collection (ATCC, CL173, Manassas, VA, USA). The cells were cultured in DMEM containing 10% bovine calf serum (BCS) and 1% antibiotics using a CO_2_ (5%) incubator at 37 °C. Differentiation of 3T3-L1 cells was initiated by a 3-day treatment with differentiation medium (0.5 mM IBMX, 1 μM DEX, and 5 μg/mL Insulin) in DMEM containing 10% FBS. To evaluate the effects of nobiletin, confluent cells were cultured in this differentiation medium containing various concentrations of nobiletin (6, 12.5, 25, and 50 μΜ) for 7 days. Cell viability assays confirmed that these doses, including the highest dose of 50 μM, did not induce cytotoxicity. This is consistent with earlier studies reporting the safe and effective use of nobiletin at 25–100 μM in adipocyte models [27,29,30]. Undifferentiated cells served as the negative control, while cells differentiated without nobiletin treatment served as the positive control.

### 2.4. Lipid Quantification

After differentiation, the cells were washed with phosphate-buffered saline (PBS) and fixed with 10% formalin for 5 min at room temperature (RT). Following fixation, the cells were rinsed with 60% isopropanol and allowed to air-dry. To stain intracellular lipid droplets, the cells were incubated with an Oil Red O working solution for 30 min at RT. Excess stain was then removed by washing the wells three times with PBS. Stained cells were visualized and imaged using a Nikon Eclipse inverted light microscope (Shinagawa, Tokyo, Japan). For quantification, the incorporated dye was eluted with 100% isopropanol, and the absorbance was measured at 500 nm. Lipid accumulation was expressed as a percentage relative to the control.

### 2.5. Quantification of Gene Expression

For gene expression analysis, total RNA was isolated using an RNA extraction kit (iNtRON Biotechnology, Seongnam-si, Republic of Korea). The purity and concentration of the isolated RNA were determined spectrophotometrically. Complementary DNA (cDNA) was then synthesized from 50 ng of total RNA using a reverse transcription kit (TaKaRa Bio, Kusatsu, Japan). Real-time quantitative PCR (RT-qPCR) was conducted using TB Green (TaKaRa Bio, Kusatsu, Japan) on an ABI QuantStudio 3 System (Applied Biosystems, Foster City, CA, USA). The expression levels of target genes were normalized to β-actin, an internal control. All reactions were performed in triplicate, and the primer sequences used for amplification are listed in Table 1.

### 2.6. Protein Quantification and Immunoblot Analysis

To analyze protein expression, control cells and nobiletin-treated cells were harvested. Total proteins were extracted using a lysis buffer supplemented with phosphatase and protease inhibitors (Thermo Fisher, Waltham, MA, USA) and incubated on ice for 30 min. The protein concentration of the resulting lysates was determined using a protein assay kit. Equal amounts of protein from each sample were separated by sodium dodecyl-sulfate polyacrylamide gel electrophoresis (SDS-PAGE) and subsequently transferred to polyvinylidene fluoride (PVDF) membranes. The membranes were blocked for 1 h at room temperature, followed by incubation with primary antibodies for 2 h. After washing, membranes were incubated with horseradish peroxidase (HRP)-conjugated secondary antibodies for 1 h. Protein bands were visualized using an enhanced chemiluminescence (ECL) reagent (Amersham Pharmacia, Piscataway, NJ, USA) and imaged with a Quant LAS 500 system (GE Healthcare, Waukesha, WI, USA). The expression levels of PPARγ, SREBP-1c, C/EBPα, FAS, UCP-1, PRDM16, PGC-1α, AMPK, and ACC were analyzed, with β-actin serving as a loading control.

### 2.7. Exosome Purification and Exosomal Marker Study

To collect exosomes secreted from differentiated adipocytes, the cells were washed with PBS 7 days post-differentiation, and the culture medium was replaced with exosome-free medium. After a 24 h incubation period, the conditioned medium was harvested. To remove cells and large debris, the harvested supernatant was subjected to a low-speed centrifugation step (3000× *g* for 30 min at 4 °C). The resulting cell-free supernatant was then stored at −80 °C until exosome isolation. Extracellular vesicles (EVs), including exosomes, were isolated from conditioned media using the XENO-EVIMEDI Kit (Cell to Bio, Seongnam-si, Republic of Korea) according to the manufacturer’s protocol. The size distribution and concentration of the isolated EVs were determined by Nanoparticle Tracking Analysis (NTA) using a ZetaView^®^ PMX 110 instrument (Particle Metrix, Meerbusch, Germany). To confirm their identity as exosomes, the presence of canonical exosomal markers (CD9, CD63, and CD81) was verified by Western Blot analysis.

### 2.8. microRNA RNA-Seq Analysis

For small RNA sequencing, adipocytes were first differentiated for 7 days. Following differentiation, the cells were washed with PBS and cultured in exosome-free medium for 24 h. Exosomes were isolated from the harvested supernatants via centrifugation using the XENO-EVIMEDI Kit (Cell to Bio, Seongnam-si, Republic of Korea). Next, total RNA was extracted from the isolated exosomes using an RNA extraction kit (iNtRON Biotechnology, Seongnam-si, Republic of Korea), and sequencing libraries were prepared using the XENO QIA Library Kit. The resulting libraries were sequenced on an Illumina MiniSeq platform to generate single-end reads. The resulting raw reads were processed through a bioinformatics pipeline. This process involved an initial quality assessment with FastQC (v0.11.5) and adapter trimming with Skewer (v0.2.2). The cleaned reads were then used to quantify miRNA expression levels with the QuickMIRSeq pipeline. Finally, all identified miRNAs were annotated against the miRBase database for final analysis.

### 2.9. Validation of microRNA

To validate the expression of specific miRNAs, exosomes were isolated from the conditioned media of differentiated 3T3-L1 cells using the XENO-EVIMEDI Kit (Cell to Bio, Seongnam-si, Republic of Korea). Total RNA was subsequently extracted from exosomes with an RNA extraction kit (iNtRON Biotechnology, Seongnam-si, Republic of Korea). The expression levels of target miRNAs were then quantified by real-time quantitative PCR (RT-qPCR) on a QuantStudio 3 system (Thermo Fisher, Waltham, MA, USA) using TB Green Master Mix (TaKaRa Bio, Kusatsu, Japan) according to the manufacturer’s protocol. All reactions were performed in triplicate. The relative expression of each miRNA was calculated after normalization to the internal control, U6 small nuclear RNA (snRNA). All primer sequences used for amplification are listed in Table 2.

### 2.10. Statistical Analysis

All experiments were performed in triplicate, and the data are presented as the mean ± standard deviation (SD). Statistical analysis was conducted using GraphPad Prism v9.5.1 (GraphPad Software, San Diego, CA, USA). Differences between groups were assessed using a one-way analysis of variance (ANOVA) followed by Tukey’s post hoc test. A *p*-value of less than 0.05 (*p* < 0.05) was considered statistically significant.

## 3. Results

### 3.1. Nobiletin Concentration Affecting the Survival of 3T3-L1 Adipocytes

To establish a non-toxic working concentration, the cytotoxicity of nobiletin on 3T3-L1 preadipocytes was evaluated. Nobiletin did not exhibit significant toxicity at concentrations up to 50 μM, as cell viability remained comparable to that of the untreated control group. Based on these results, a maximum concentration of 50 μM was used for all subsequent experiments. This finding is consistent with previous studies reporting the use of similar or even higher concentrations of nobiletin without adverse effects on cell viability [30,31].

### 3.2. Reduction of Intracellular Lipid Accumulation in 3T3-L1 Adipocytes in the Presence of Nobiletin

To determine the effect of nobiletin on adipogenesis, intracellular lipid accumulation was assessed by Oil Red O staining 7 days after the induction of differentiation. As expected, the positive control cells (differentiated in the absence of nobiletin) exhibited robust intracellular lipid droplet formation (Figure 1a). However, cells treated with nobiletin displayed a significant, concentration-dependent decrease in lipid content (Figure 1b). These results demonstrate that nobiletin impairs adipogenesis in 3T3-L1 cells.

### 3.3. Suppression of Adipogenic Transcription Factors by Nobiletin

To understand the mechanism behind nobiletin’s anti-adipogenic effect, we investigated its influence on the master transcriptional regulators of adipogenesis: PPARγ, SREBP-1c, and C/EBPα. RT-PCR analysis showed that nobiletin significantly downregulated the mRNA expression of *srebp-1c* and *c/ebpα* at all tested concentrations. The expression of *pparγ* mRNA was also reduced, but only at the higher concentrations of 25 and 50 μM. In agreement with these findings, Western Blot analysis confirmed that nobiletin treatment caused a significant and dose-dependent suppression of the protein levels for all three transcription factors (Figure 2).

### 3.4. Inhibition of the Fatty Acid Synthesis Enzyme by Nobiletin

To further elucidate nobiletin’s anti-adipogenic mechanism, we analyzed the expression of Fatty Acid Synthase (FAS), a critical enzyme for fatty acid synthesis [32]. Nobiletin markedly suppressed FAS expression in a dose-dependent manner. Both RT-PCR and Western Blot analyses confirmed a significant reduction in FAS at the transcript and protein levels (Figure 2d,h), directly correlating with the observed decrease in intracellular lipid storage.

### 3.5. Increase in the Expression of Browning Markers in Differentiated Adipocytes 3T3-L1 in the Presence of Nobiletin

To investigate whether nobiletin induces adipocyte browning, we examined the expression of core thermogenic markers, including Uncoupling Protein 1 (UCP-1) [33], PRDM16 [34], and PGC-1α [35]. The expression of these key browning-related factors was assessed at both the transcript and protein levels. At the transcript level, nobiletin significantly enhanced the expression of all browning-related genes relative to the control, with *prdm16* and *pgc-1α* peaking at 50 μM (Figure 3a–c). In parallel, Western Blot analysis confirmed that protein levels of UCP-1, PRDM16, and PGC-1α were also dose-dependently upregulated by nobiletin (Figure 3d–f).

### 3.6. Activation of AMPK and ACC by Nobiletin in 3T3-L1 Cells

To investigate the anti-adipogenic mechanism of nobiletin, we analyzed AMPK activation in adipocytes. Treatment with nobiletin at 6, 12.5, and 25 μM led to a significant, dose-dependent increase in the ratio of phosphorylated AMPK (Figure 3g). Consequently, the phosphorylation of ACC, a downstream target of AMPK, also increased in a dose-dependent manner (Figure 3h). These results support the hypothesis that nobiletin’s activation of AMPK leads to the phosphorylation and inhibition of ACC, thereby suppressing lipid accumulation by inhibiting fatty acid synthesis and promoting fatty acid oxidation.

### 3.7. Exosome Extraction, Size, Concentration, and Markers

Following a 7-day differentiation period, the adipocytes were washed with PBS and incubated in exosome-free medium for 24 h to collect conditioned media. The harvested supernatants were then centrifuged, and the isolated exosomes were characterized using a ZetaView particle analyzer (Particle Metrix GmbH, Inning am Ammersee, Germany). Nanoparticle Tracking Analysis (NTA) confirmed the successful isolation of exosomes, with median diameters of 160.2 nm (undifferentiated, E), 139.5 nm (differentiated, EN0), and 160.3 nm (nobiletin-treated, EN50) (Figure 4a). The concentrations were found to be 2.6 × 10^7^, 6.3 × 10^7^, and 1.1 × 10^8^ particles/mL for the E, EN0, and EN50 groups, respectively. To validate their identity and purity, Western Blot analysis was performed. All isolated vesicles were positive for the canonical exosomal markers CD9, CD63, and CD81. Conversely, the cytosolic protein β-actin, a negative control for exosome purity, was not detected, indicating that the preparations were free from cellular contamination (Figure 4b).

### 3.8. Expression Analysis of microRNA

To profile the miRNA content of the exosomes, we performed small RNA sequencing. The analysis successfully quantified 235 distinct miRNAs across all samples (Appendix A). We then performed differential expression analysis to identify miRNAs altered by differentiation (EN0 vs. E) and by subsequent nobiletin treatment (EN50 vs. EN0), using a log2 fold-change cutoff. During differentiation (EN0 vs. E), 98 miRNAs were significantly upregulated, while 31 were downregulated. In contrast, nobiletin treatment (EN50 vs. EN0) resulted in the upregulation of 11 miRNAs and the downregulation of 32 miRNAs (Appendix A). To identify miRNAs whose expression was specifically reversed by nobiletin, we performed a Venn analysis. This approach identified a key subset of 4 miRNAs that were downregulated during differentiation but upregulated following nobiletin treatment. A second key finding was a group of 21 miRNAs induced during adipocyte differentiation (E vs. EN0) but then suppressed following nobiletin treatment (EN0 vs. EN50) (Figure 5). The expression signature of this group is particularly noteworthy, as many of these miRNAs have established roles in promoting adipogenesis. Specifically, the nobiletin-induced downregulation of eight of these pro-adipogenic miRNAs directly aligns with the observed reduction in lipid accumulation and is supported by existing literature (Figure 6) [36,37,38,39,40,41,42,43,44,45].

### 3.9. Validation of Expressed microRNAs

To validate the findings from our RNA-Seq analysis, we selected ten key miRNAs implicated in lipid metabolism for confirmation by RT-qPCR. The study confirmed that nobiletin treatment led to significant upregulation of miR-181d-5p and miR-221-3p compared with the untreated control. Conversely, the remaining eight miRNAs (miR-205-5p, miR-331-3p, miR-130b-3p, miR-143-5p, miR-183-3p, miR-196b-5p, miR-26b-3p, and miR-378d) were all significantly downregulated (Figure 7). Crucially, the expression patterns of all ten miRNAs as determined by RT-qPCR were in strong agreement with the RNA-Seq data. This concordance validates the reliability of our high-throughput sequencing analysis.

## 4. Discussion

This study confirms nobiletin’s anti-adipogenic potential by demonstrating its ability to inhibit lipid accumulation in 3T3-L1 adipocytes. This finding aligns with most previous reports [31,46,47] and supports in vivo evidence of nobiletin’s anti-obesity effects [25,48,49], despite a few contrasting studies that reported a pro-adipogenic role [29,50]. To elucidate the underlying mechanism, we investigated nobiletin’s impact on the master regulators of adipogenesis. Our results revealed that nobiletin treatment significantly suppressed the expression of the key transcription factors PPARγ, C/EBPα, and SREBP-1c. This finding strongly suggests that the downregulation of this core transcriptional network is a key molecular mechanism driving the anti-adipogenic effects of nobiletin. Our findings suggest that nobiletin remodels adipocyte metabolism through a powerful dual mechanism: simultaneously inhibiting fat storage while promoting fat burning. The inhibition of fat storage is evidenced by the suppression of SREBP-1c and its downstream target, Fatty Acid Synthase (FAS), a critical enzyme for de novo lipogenesis [51,52]. In parallel, nobiletin promotes a thermogenic program, as evidenced by robust upregulation of the entire PRDM16/PGC-1α/UCP-1 browning [34].

Crucially, these two distinct metabolic arms appear to be orchestrated by a single upstream regulator: AMP-activated protein kinase (AMPK). In support of this unifying mechanism, our analysis confirmed that nobiletin significantly enhances AMPK phosphorylation, a finding consistent with previous reports [25,27]. Future studies using AMPK inhibitors are needed to confirm this pathway. Activated AMPK is known to perform both functions observed in our study: it directly phosphorylates and inactivates ACC, the rate-limiting enzyme in fatty acid synthesis, and promotes the PGC-1α-mediated browning cascade. Therefore, AMPK activation emerges as the central node through which nobiletin exerts its potent anti-adipogenic effects.

Beyond its direct effects on cellular signaling, this study provides the first evidence that nobiletin remodels adipocyte exosomal miRNA profiles, revealing a novel layer of regulation. Our analysis focused on a key subset of four miRNAs whose expression was downregulated during differentiation but subsequently rescued or upregulated by nobiletin treatment.

Notably, two of these miRNAs, miR-181d-5p and miR-221-3p, have been shown to have anti-adipogenic roles. For instance, miR-181d-5p is known to target ANGPTL3, a key player in lipid metabolism and adipocyte differentiation [37]. Similarly, the overexpression of miR-221-3p has been shown to reduce lipid storage in human adipocytes [36]. Crucially, the expression of miR-221-3p is negatively correlated with lipid metabolism genes, such as FAS, which aligns perfectly with our observation that nobiletin suppresses FAS expression. Therefore, the nobiletin-induced upregulation of these specific anti-adipogenic miRNAs represents a novel, indirect mechanism that contributes to its overall anti-obesity effect [36].

Conversely, our analysis identified a second, larger cohort of 21 miRNAs with the opposite expression pattern: they were upregulated during differentiation and subsequently suppressed by nobiletin treatment. The nobiletin-induced suppression of this group is particularly significant, as many of these miRNAs are known to be pro-adipogenic, meaning they actively promote fat cell development and lipid storage [36,37,38,39,40,41,42,43,45]. For instance, miR-331-3p is known to enhance fat storage and fatty acid synthesis [39]. Another key example is miR-205-5p, which promotes adiposity by repressing the anti-adipogenic factor ZEB1. By downregulating miR-205-5p, nobiletin likely releases the inhibition on ZEB1, thereby contributing to the suppression of adiposity. Therefore, nobiletin’s ability to dismantle this network of pro-adipogenic miRNAs represents another powerful mechanism underlying its anti-obesity effects. Several of these miRNAs are known to directly promote fat storage. For example, miR-183 enhances lipid accumulation by upregulating master adipogenic genes, such as PPARγ and SREBP-1c, while miR-331-3p drives fatty acid synthesis [40]. Concurrently, nobiletin also appears to promote a thermogenic phenotype by downregulating miRNAs known to inhibit browning. For instance, miR-130b-3p is a direct inhibitor of the key browning factor PGC-1α [40]. Similarly, miR-143 has been shown to suppress a signaling pathway essential for adipose tissue browning [53]. Therefore, the nobiletin-induced downregulation of this miRNA network provides a sophisticated mechanism that both dismantles the pathways for fat storage and removes the brakes on energy expenditure [42].

The significance of this pro-adipogenic miRNA network is further highlighted by the specific roles of other downregulated members, revealing a multi-pronged molecular strategy. For example, our suppression of miR-183 is consistent with the observed downregulation of master adipogenic genes (PPARγ, C/EBPα, SREBP-1c), as miR-183 is a known positive regulator of this network (Figure 6). Similarly, nobiletin’s suppression of miR-196b-5p and miR-26b is expected to interfere with pro-lipogenic signaling pathways. The latter is particularly compelling, as miR-26b normally inhibits the anti-adipogenic factor COX-2; thus, by downregulating miR-26b, nobiletin likely unleashes COX-2 activity to suppress fat accumulation [54,55]. Furthermore, the suppression of miR-196b-5p and miR-378d interferes with pro-lipogenic mTORC1/TGF-β signaling and releases the brakes on Nrf1, a repressor of PPARγ, respectively [56]. Thus, nobiletin’s ability to coordinately downregulate this diverse suite of miRNAs, each with distinct pro-obesity functions, underscores the robustness of its anti-adipogenic action.

Although our results indicate that AMPK activation and exosomal miRNA regulation contribute to the anti-adipogenic and browning-inducing effects of nobiletin, the exact relationship between these two mechanisms remains to be elucidated. Since it is unclear whether their actions are synergistic, sequential, or independent, further studies are needed to determine the potential interactions or interdependencies between miRNA networks and AMPK signaling.

In summary, this study demonstrates that nobiletin modulates a significant shift in the exosomal miRNA profile, contributing to its anti-obesity effects by upregulating key anti-adipogenic miRNAs while suppressing a broad network of pro-adipogenic ones. The strong concordance between our RNA-Seq and RT-qPCR data for all ten selected miRNAs validates the accuracy of these findings. This work not only uncovers a novel layer of nobiletin’s mechanism but also identifies these validated miRNAs as promising therapeutic targets for future research into obesity.

## 5. Conclusions

The current study demonstrates that nobiletin potently suppresses adipogenesis in 3T3-L1 cells by modulating a multi-layered molecular response. We have shown that nobiletin simultaneously dismantles the core adipogenic program by downregulating master transcription factors (PPARγ, C/EBPα, SREBP-1c) while promoting a thermogenic phenotype by upregulating key browning markers. Furthermore, this study uncovers a novel regulatory layer to this mechanism, revealing for the first time that nobiletin modulates the exosomal secretome. We identified and validated a signature of ten key miRNAs whose expression patterns—upregulating anti-adipogenic miRNAs while suppressing pro-adipogenic ones—strongly align with the observed cellular effects and previously reported functions. Collectively, these findings provide a comprehensive molecular basis for nobiletin’s anti-obesity action and validate a novel set of exosomal miRNAs as promising therapeutic targets for future development.

## Figures and Tables

**Figure 1 cimb-48-00036-f001:**
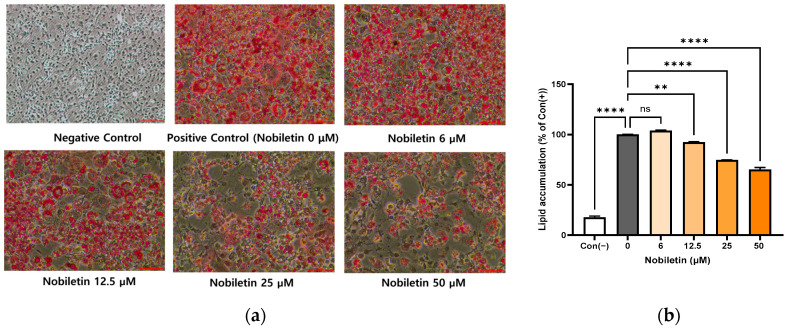
Reduction in intracellular lipid accumulation after treatment with nobiletin at different concentrations. (**a**) Quantified lipid accumulation after 7 days of nobiletin treatment during the differentiation of 3T3-L1 cells. The scale bar is 100 μm and was automatically generated by the microscope software (NIS-Elements Viewer version 4.5). (**b**) Visualized lipid droplets after treatment in the absence or presence of nobiletin (6, 12.5, 25, and 50 μM). The data are presented as mean ± SD for more than three experiments. ns, not significant; ** *p* < 0.01 and **** *p* < 0.001 vs. 0.

**Figure 2 cimb-48-00036-f002:**
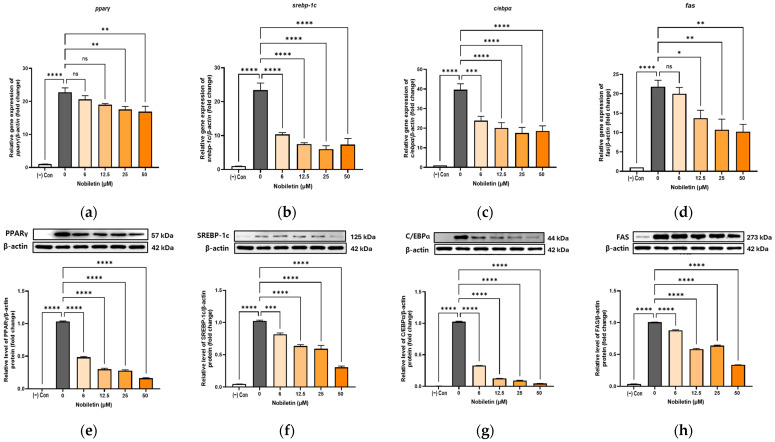
Reduction in gene and protein expression of adipogenic transcription factors after treatment with nobiletin. The mRNA expression of *pparγ* (**a**), *srebp-1c* (**b**), *c/ebpα* (**c**), and *fas* (**d**) was measured by real-time PCR, normalized by *β-actin*. Protein expressions of PPARγ (**e**), SREBP-1c (**f**), C/EBPα (**g**), and FAS (**h**) were measured by Western Blotting, normalized by β-actin. The data are presented as mean ± SD for more than three experiments. ns, not significant; * *p* < 0.05, ** *p* < 0.01, *** *p* < 0.001, and **** *p* < 0.001 vs. 0.

**Figure 3 cimb-48-00036-f003:**
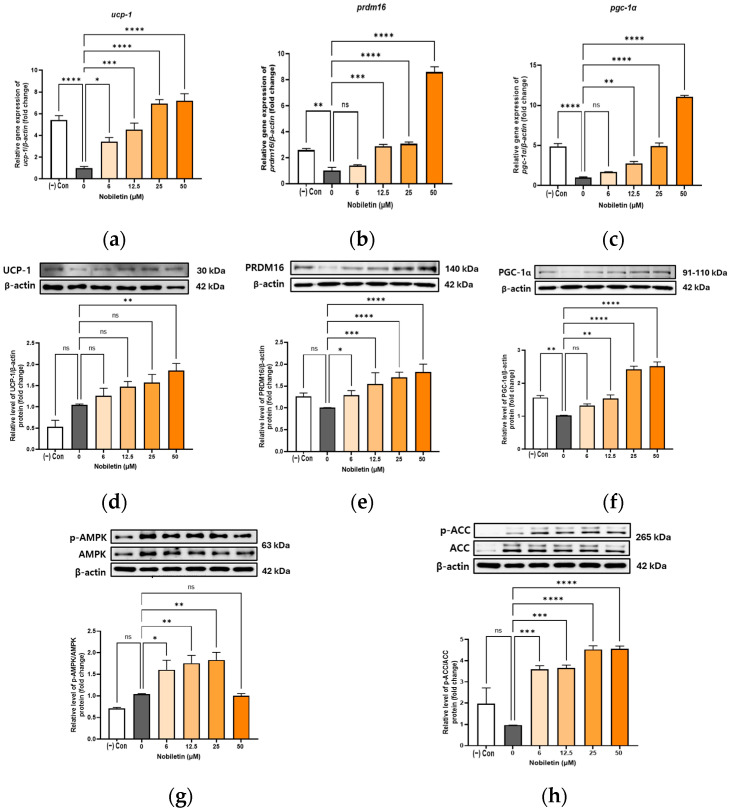
Differences in protein and gene expression of selected genes after nobiletin treatment. The mRNA expression levels of *ucp-1* (**a**), *prdm16* (**b**), and *pgc-1α* (**c**) were measured by real-time PCR, normalized to *β-actin*. Protein expressions of UCP-1 (**d**), PRDM16 (**e**), PGC-1α (**f**), AMPK (**g**), and ACC (**h**) were measured by Western Blotting and normalized to β-actin. The data are presented as mean ± SD for more than three experiments. ns, not significant; * *p* < 0.05, ** *p* < 0.01, *** *p* < 0.001, and **** *p* < 0.001 vs. 0.

**Figure 4 cimb-48-00036-f004:**
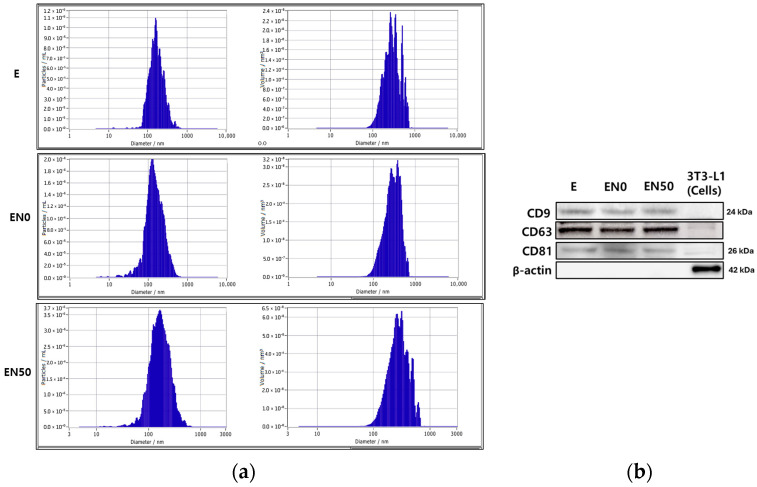
Characterization and validation of isolated exosomes. (**a**) Nanoparticle tracking analysis (NTA) of the size distribution and concentration of exosomes isolated from different treatment groups. Groups are defined as: E, undifferentiated 3T3-L1 cells (negative control); EN0, differentiated 3T3-L1 adipocytes (positive control); and EN50, differentiated adipocytes treated with 50 μM nobiletin. (**b**) Western Blot analysis confirming the presence of exosome marker proteins (CD9, CD63, and CD81) in the isolated vesicles.

**Figure 5 cimb-48-00036-f005:**
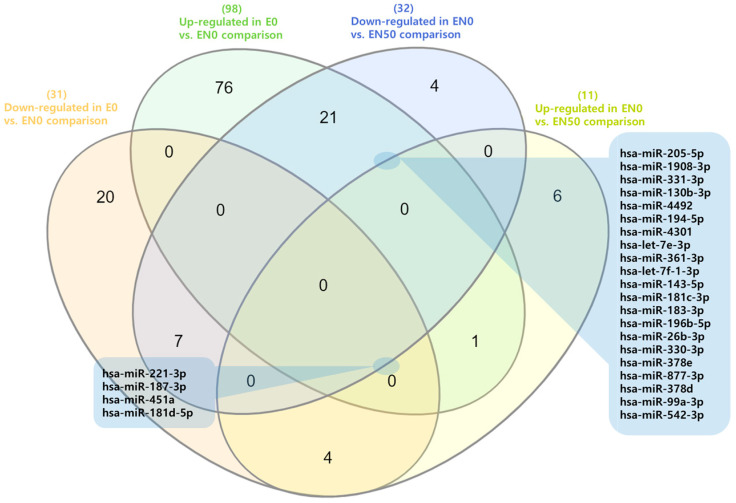
The Venn diagram shows the selected common miRNAs.

**Figure 6 cimb-48-00036-f006:**
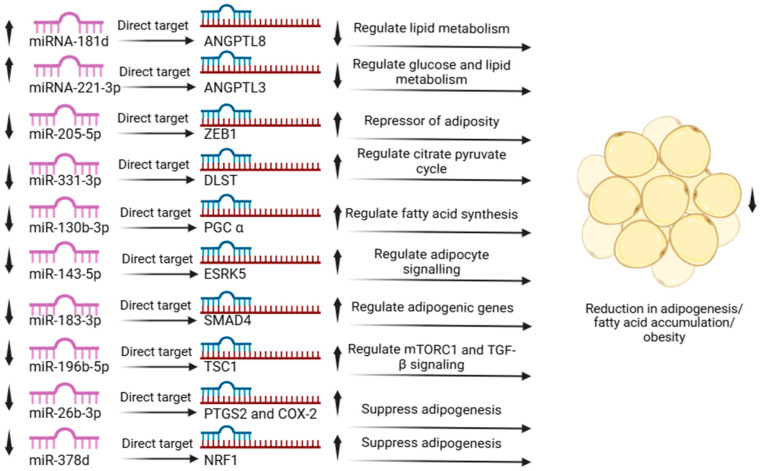
Differentially expressed miRNA with target and essential role in adipogenesis/obesity. up arrow denotes up-regulation, and the down arrow denotes suppression.

**Figure 7 cimb-48-00036-f007:**
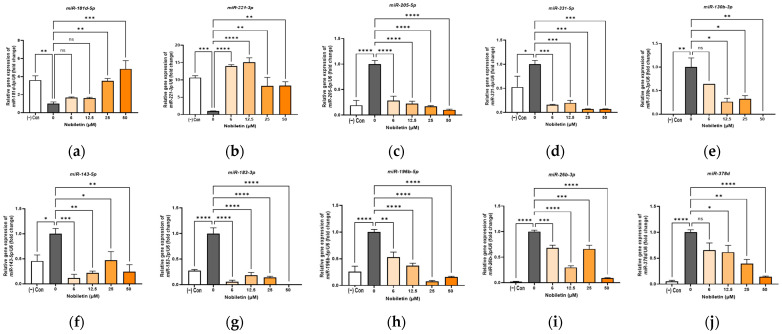
Expression of miRNAs involved in adipogenesis and obesity. The relative expression levels of (**a**) miR-181d, (**b**) miR-221-3p, (**c**) miR-205-5p, (**d**) miR-331-3p, (**e**) miR-130b-3p, (**f**) miR-143-5p, (**g**) miR-183-3p, (**h**) miR-196b-5p, (**i**) miR-26b-3p, and (**j**) miR-378d were quantified. ns, not significant; * *p* < 0.05, ** *p* < 0.01, *** *p* < 0.001, and **** *p* < 0.001 vs. 0.

**Table 1 cimb-48-00036-t001:** Primers used in quantitative real-time PCR for gene expression.

Gene Name	Forward (5′-3′)	Reverse (5′-3′)
*PPARγ*	TTTTCAAGGGTGCCAGTTTC	AATCCTTGGCCCTCTGAGAT
*C/EBPα*	TTACAACAGGCCAGGTTTCC	GGCTGGCGACATACAGTACA
*SREBP-1c*	TGTTGGCATCCTGCTATCTG	AGGGAAAGCTTTGGGGTCTA
*FAS*	TTGCTGGCACTACAGAATGC	AACAGCCTCAGAGCGACAAT
*PGC-1α*	GCAACATGCTCAAGCCAAAC	TGCAGTTCCAGAGAGTTCCA
*UCP-1*	CTTTGCCTCACTCAGGATTGG	ACTGCCACACCTCCAGTCATT
*PRDM16*	CAGCACGGTGAAGCCATTC	GCGTGCATCCGCTTGTG
*β-actin*	CTGTCCCTGTATGCCTCTG	ATGTCACGCACGATTTCC

*PPARγ*: peroxisome proliferator-activated receptor gamma, *C/EBPα*: CCAAT/enhancer-binding protein alpha, *SREBP-1c*: Sterol regulatory element-binding transcription factor-1, *FAS*, fatty acid synthase, *PGC-1α*: Peroxisome proliferator-activated receptor gamma coactivator 1-alpha, *UCP-1*: Uncoupling protein 1, *PRDM16*: PR domain-containing 16, *β-actin*: Beta-actin (internal loading control).

**Table 2 cimb-48-00036-t002:** Primers used in microRNA quantitative real-time PCR.

microRNA	Forward (5′-3′)	Reverse (5′-3′)
miR-221-3p	GCAGAGCTACATTGTCTGCT	CAGTTTTTTTTTTTTTTTGAAACCCA
miR-181d	ACACTCCAGCTGGGAACATTCATTGTTGTC	TCAACTGGTGTCGTGGAGTCGGCAATTCAG-TTGAGACCCACCG
miR-205-5p	TCCTTCATTCCACCGGAGTCTG	GCGAGCACAGAATTAATACGAC
miR-331-3p	ACACTCCAGCTGGGGCCCCTGGGCCTATC	CTCAACTGGTGTCGTGGAGTCGGCAATTCAGTTGAGTTCTAGGA
miR-130b-3p	CAGTGCAATGATGAAAGGGCAT	ATGCCCTTTCATCATTGCACTG
miR-143-5p	GCGCAGCGCCCTGTCTCC	GCTGCAGAACAACTTCTC
miR-183-3p	CGCAGAGTGTGACTCCTGTT	TGGCCCTTCGGTAATTCACT
miR-196b-5p	ACACTCCAGCTGGGTAGGTAGTTTCCTGTT	CTCAACTGGTGTCGTGGAGTCGGCAATTCA-GTTGAGCCCAACAA
miR-26b-3p	GGCCTGTTCTCCATTACTTGG	CGCTTCACGAATTTGCGTGTCAT
miR-378	GGACACTGGACTTGGAG	TGCGTGTCGTGGAGTC
U6	CTCGCTTCGGCAGCACA	ACGCTTCACGAATTTGCGT

## Data Availability

The original contributions presented in this study are included in the article/Appendix A. Further inquiries can be directed to the corresponding authors.

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
