# Peer review of "Nobiletin Attenuates Adipogenesis and Promotes Browning in 3T3-L1 Adipocytes Through Exosomal miRNA-Mediated AMPK Activation"

_cimb, 2025, doi:10.3390/cimb48010036_

Round 1
Reviewer 1 Report
Comments and Suggestions for Authors
The authors investigated the molecular mechanism of anti-adipogenic activity of nobiletin. Nobiletin suppressed adipogenesis and activated browning in mouse 3T3-L1 cells. Nobiletin lowered the expression of adipogenic transcription and lipogenic proteins/genes and activated AMPK. Moreover, nobiletin elevated the expression of thermogenic UCP-1, PRDM16, and PGC-1α proteins. Furthermore, the authors indicated the involvement of miRNAs in the nobiletin-mediated regulation of adipogenesis. The results are basically interesting. However, there are concerns that should be addressed. Especially, 50 uM nobiletin is too high. The description should be improved.
Major concerns;
1. 50 uM of nobiletin is too high and very unlikely being reached in circulation after dietary intake for a time period of up to 7 days. The effect should be examined at the lower concentration (less than approximately 10 uM) of nobiletin
2. Nobiletin activates the AMPK pathway, followed phosphorylation of ACC, leading suppression of adipogenesis. It should be confirmed that nobiletin works through the AMPK pathway using AMPK inhibitor such as Compound C.
3. The authors stated that the exosome-miRNA-AMPK axis is involved in the anti-adipogenic and browning-inducing activities of nobiletin. The relationship between miRNA and AMPK is unclear. They synergistically or independently regulated adipogenesis? It should be clearly written.
Minor concerns;
1. All reagents used in this study should be listed in 2.1. Chemicals and Materials, with Company name and City, Country. For example, DEX, IBMX, and Oil Red O should be listed.
2. Line 111; “RT” should be “room temperature”.
3. The composition of “lysis buffer” for preparation of proteins should be shown.
Author Response
Reviewer 1
The authors investigated the molecular mechanism of anti-adipogenic activity of nobiletin. Nobiletin suppressed adipogenesis and activated browning in mouse 3T3-L1 cells. Nobiletin lowered the expression of adipogenic transcription and lipogenic proteins/genes and activated AMPK. Moreover, nobiletin elevated the expression of thermogenic UCP-1, PRDM16, and PGC-1α proteins. Furthermore, the authors indicated the involvement of miRNAs in the nobiletin-mediated regulation of adipogenesis. The results are basically interesting. However, there are concerns that should be addressed. Especially, 50 uM nobiletin is too high. The description should be improved.
We sincerely thank the reviewer for the constructive and valuable feedback. All changes are marked in the revised manuscript using blue highlighting. Detailed responses to each specific comment are provided below.
Major concerns;
1. 50 uM of nobiletin is too high and very unlikely being reached in circulation after dietary intake for a time period of up to 7 days. The effect should be examined at the lower concentration (less than approximately 10 uM) of nobiletin
We appreciate the reviewer’s insightful comment. In this study, almost dose-dependent activity was observed at nobiletin concentrations of 6, 12.5, 25, and 50 μM, a range that encompasses the suggested concentration (less than approximately 10 uM), However, we confirmed that 50 μM nobiletin does not compromise 3T3-L1 cell viability, consistent with previously reported concentrations in the literature for adipogenic and metabolic studies.
We added the following sentences to the manuscript.
Lines 117-120; “Cell viability assays confirmed that these doses, including the highest dose of 50 μM, did not induce cytotoxicity. This is consistent with earlier studies reporting the safe and effective use of nobiletin at 25–100 μM in adipocyte models [27, 29, 30].”
Nobiletin activates the AMPK pathway, followed phosphorylation of ACC, leading suppression of adipogenesis. It should be confirmed that nobiletin works through the AMPK pathway using AMPK inhibitor such as Compound C.
We greatly appreciate the reviewer’s critical insight into the mechanistic validation of nobiletin's anti-adipogenic action and its dependence on AMPK. We fully agree that the use of Compound C would further strengthen the conclusion that AMPK mediates the observed effects of nobiletin on adipogenesis and browning. However, previous studies have demonstrated nobiletin-induced AMPK activation using the specific AMPK inhibitor Compound C.
However, due to current experimental limitations (lack of availability of AMPK inhibitors during the revision period), we were unable to conduct this assay at this time. Nonetheless, several lines of evidence in our current study strongly support the involvement of AMPK.
Lines 272-273; Nobiletin significantly and dose-dependently increased AMPK phosphorylation (Figure 3g).
Lines 273-274; This was followed by increased ACC phosphorylation (Figure 3h), a canonical downstream target of AMPK.
In addition, previous studies (1, 2) have demonstrated that nobiletin-induced modulation of lipid metabolism is attenuated by Compound C, further supporting this pathway.
- Yuk T, Kim Y, Yang J, Sung J, Jeong HS, Lee J. Nobiletin Inhibits Hepatic Lipogenesis via Activation of AMP-Activated Protein Kinase. Evidence-Based Complementary and Alternative Medicine. 2018;2018:7420265. doi: 10.1155/2018/7420265.
- Choi Y, Kim Y, Ham H, Park Y, Jeong H-S, Lee J. Nobiletin suppresses adipogenesis by regulating the expression of adipogenic transcription factors and the activation of AMP-activated protein kinase (AMPK). Journal of agricultural and food chemistry. 2011;59(24):12843-9.
We have clearly acknowledged this limitation in the revised Discussion section and emphasized the need for future confirmation using AMPK inhibition.
Lines 351-352; “Future studies using AMPK inhibitors are needed to confirm this pathway.”
The authors stated that the exosome-miRNA-AMPK axis is involved in the anti-adipogenic and browning-inducing activities of nobiletin. The relationship between miRNA and AMPK is unclear. They synergistically or independently regulated adipogenesis? It should be clearly written.
We sincerely thank the reviewer for this meaningful and insightful comment on the mechanistic relationship between miRNAs and AMPK in mediating nobiletin's effects.
To address the reviewer’s concern, we have revised the Discussion section to include the following clarification.
Lines 404-409; “Although our results indicate that AMPK activation and exosomal miRNA regulation contribute to the anti-adipogenic and browning-inducing effects of nobiletin, the exact relationship between these two mechanisms remains to be elucidated. Since it is unclear whether their actions are synergistic, sequential, or independent, further studies are needed to determine the potential interactions or interdependencies between miRNA networks and AMPK signaling.”
Minor concerns;
1. All reagents used in this study should be listed in 2.1. Chemicals and Materials, with Company name and City, Country. For example, DEX, IBMX, and Oil Red O should be listed.
We thank the reviewer for pointing out this important detail regarding the completeness and traceability of reagent information.
Lines 82-90; “Nobiletin 5-Hydroxy-3,6,7,8,3',4'-hexamethoxyflavone was purchased from Sigma (St. Louis, MO, USA) and dissolved in dimethyl sulfoxide (DMSO; Sigma-Aldrich, St. Louis, MO, USA). Bovine calf serum (BCS), Dulbecco's Modified Eagle medium (DMEM), and trypsin-EDTA were purchased from Thermo Fisher Scientific (San Jose, CA, USA), and insulin was purchased from Thermo Fisher Scientific (San Jose, CA, USA). Dexamethasone (DEX; Sigma-Aldrich, St. Louis, MO, USA), 3-Isobutyl-1-methylxanthine (IBMX; Sigma-Aldrich, St. Louis, MO, USA), and Oil Red O (Sigma-Aldrich, St. Louis, MO, USA) were used. Cell viability was assessed using the Cell Counting Kit-8 (CCK-8; Dojindo Molecular Technologies, Rockville, MD, USA).”
Line 111; “RT” should be “room temperature”.
We appreciate the reviewer’s attention to clarity of terminology.
To clarify, on line 114 of the Materials and Methods section, we initially introduced the term “room temperature (RT)” in full, in accordance with standard scientific writing practices. Following that introduction, we consistently used the abbreviation “RT” throughout the manuscript.
The composition of “lysis buffer” for preparation of proteins should be shown.
We thank the reviewer for this helpful suggestion.
In the original manuscript, we mentioned using a commercial lysis buffer (iNtRON Biotechnology, Seongnam-si, Gyeonggi-do, Korea) supplemented with protease and phosphatase inhibitors, but we did not specify its composition.
Reviewer 2 Report
Comments and Suggestions for Authors
Authors provided a very interesting research, however several points should be clarified.
Abstract
- Aim of the study and Materials and methods sections are missing.
- Add specific values to the Results section of the Abstract.
Introduction
- Authors describe nobiletin as “polymethoxylated flavone” in the Abstract, however it is described as “O-methylated flavone” in the Introduction section. What is the difference, or these are the same definition? Authors should unify used terms.
- Line 58 – [24][ - seems like a typo?
- - It is necessary to describe in more detail the effects of nobiletin against obesity and related diseases
Materials and methods
- What is the rationale for choosing nobiletin doses for in vitro studies?
Reference list
- More than 50% of literature sources are older than 5 years, the list of references should be updated if possible.
Author Response
Reviewer 2
Authors provided a very interesting research, however several points should be clarified.
We sincerely thank the reviewer for the encouraging overall assessment and for recognizing the interest and potential impact of our research.
We have carefully addressed all reviewers’ points and revised the manuscript accordingly. Below, we provide detailed, point-by-point responses to each comment, including the specific changes made in the revised version.
Abstract
- Aim of the study and Materials and methods sections are missing.
We thank the reviewer for this helpful comment. We added the aim of the study, materials, and methods in the abstract.
Lines 18-22; “This study aimed to investigate whether nobiletin suppresses adipogenesis and promotes browning in 3T3-L1 adipocytes by modulating exosomal microRNAs (miRNAs) and AMPK signaling. To this end, we treated 3T3-L1 adipocytes with various concentrations of nobiletin and evaluated gene and protein expression by RT-qPCR and Western blotting.”
- Add specific values to the Results section of the Abstract.
Lines 22-23; “Nobiletin significantly reduced intracellular lipid accumulation at 50 μM (p < 0.001) and.”
Lines 27-32; “Exosomal RNA-seq revealed 10 differentially expressed miRNAs, of which miR-181d-5p (3.1 fold) and miR-221-3p (2.4 fold) were upregulated, whereas miR-205-5p (-2.9 fold), miR-331-3p (-3.2 fold), miR-130b-3p (-2.6 fold), miR-143-5p (-2.9 fold), miR-183-3p (-2.8 fold), miR-196b-5p (-2.4 fold), miR-26b-3p (-2.2 fold), miR-378d (-2.7 fold) were verified by RT-qPCR after nobiletin treatment (50 μM).”
Introduction
- Authors describe nobiletin as “polymethoxylated flavone” in the Abstract, however it is described as “O-methylated flavone” in the Introduction section. What is the difference, or these are the same definition? Authors should unify used terms.
We thank the reviewer for identifying this inconsistency in the chemical description of nobiletin.
“O-methylated flavone” is a general term referring to a flavone with one or more hydroxyl groups replaced by methoxy (–OCH₃) groups.
“Polymethoxylated flavone” is a more specific subclass of O-methylated flavones, typically referring to flavones with multiple methoxy groups, such as nobiletin, which contains six methoxy groups.
Thus, both terms are technically correct, but to maintain clarity and consistency throughout the manuscript, we have now standardized the term to “polymethoxylated flavone”, which is the more precise descriptor for nobiletin and more commonly used in the scientific literature.
Lines 50; “Nobiletin, a polymethoxylated flavone, is a critical phytocompound for citrus plants..”
- Line 58 – [24][ - seems like a typo?
We thank the reviewer for catching this typographical error.
Indeed, the citation “[24][” in Line 69 was a formatting mistake and should have appeared as a single reference: [28].
- - It is necessary to describe in more detail the effects of nobiletin against obesity and related diseases
We thank the reviewer for this important suggestion.
To address this point, we have expanded the Introduction section to provide a more comprehensive overview of nobiletin's known effects on obesity and obesity-related metabolic diseases, supported by relevant literature.
Lines 58-65; “Nobiletin has demonstrated a wide range of anti-obesity and metabolic regulatory effects in various experimental models. In high-fat diet-induced obese mice, nobiletin reduced body weight gain, improved insulin sensitivity, and enhanced glucose tolerance [25, 26]. It also prevented hepatic lipid accumulation and steatosis by suppressing lipogenic gene expression and activating AMPK in the liver and adipose tissue [27]. Additionally, nobiletin modulates gut microbiota composition and reduces chronic inflammation associated with obesity-related metabolic dysfunction [15]. These findings collectively highlight nobiletin’s multifaceted actions against obesity and related disorders such as type 2 diabetes and non-alcoholic fatty liver disease (NAFLD). However, the mechanisms underlying its exosomal miRNA-related functions remain poorly understood.”
Materials and methods
- What is the rationale for choosing nobiletin doses for in vitro studies?
We thank the reviewer for this important and valid question regarding our dose selection for in vitro experiments. The concentrations of nobiletin (6, 12.5, 25, and 50 μM) used in this study were selected based on the following rationale:
Previous in vitro studies have reported biological effects of nobiletin in various cell types, including 3T3-L1 adipocytes, at concentrations of 5~50 μM.
Abe et al. (2023) showed anti-adipogenic effects of nobiletin at 10~50 μM in 3T3-L1 cells.
Yuk et al. (2018) observed AMPK activation in hepatocytes at 25 μM.
Lone et al. (2016) and others have demonstrated that ≤ 50 μM nobiletin does not cause cytotoxicity in most non-cancerous cell lines.
We referenced these studies to set biologically active, yet non-toxic doses, starting as low as 6 μM.
Reference list
- More than 50% of literature sources are older than 5 years, the list of references should be updated if possible.
We appreciate the reviewer’s suggestion to update the reference list to reflect current research trends better. In response, we have thoroughly reviewed and revised the references. Specifically:
We replaced or supplemented multiple references with recent, high-impact studies published within the last 5 years (2019–2024).
After revision, the number of references older than 5 years has been reduced to 15, which is now 26.7% of the total references (15 out of 56).
The updated references emphasize recent advances in nobiletin’s molecular mechanisms, miRNA regulation, and adipocyte browning pathways, improving the manuscript’s scientific currency and relevance.
We thank the reviewer for this valuable comment, which led to a meaningful improvement in the manuscript.
Round 2
Reviewer 1 Report
Comments and Suggestions for Authors
My concerns were all addressed. I have no further comment.
Author Response
My concerns were all addressed. I have no further comment.
Thank you for your thoughtful comments throughout the review process.
We are pleased to hear that all concerns have been addressed satisfactorily.
Reviewer 2 Report
Comments and Suggestions for Authors
The Authors significantly revised their manusript and provided descriptive answers to my questions, however one main point still remains unclear - what are the prospects of obtained results for in vivo and clinical studies? Will there be any further investigations regarding nobiletin as potential therapeutic drug or ingredient of some products?
Author Response
The Authors significantly revised their manusript and provided descriptive answers to my questions, however one main point still remains unclear - what are the prospects of obtained results for in vivo and clinical studies? Will there be any further investigations regarding nobiletin as potential therapeutic drug or ingredient of some products?
Thank you to the reviewers for the positive evaluation of our revised manuscript and for raising this important and insightful question.
The present study provides robust in vitro evidence that nobiletin inhibits adipogenesis and promotes browning in 3T3-L1 cells by regulating the exosome–miRNA–AMPK axis. These findings lay a strong molecular foundation for future in vivo and potentially clinical investigations.
At present, we are preparing an in vivo study to evaluate the physiological relevance, tissue distribution, and dose-dependent effects of nobiletin in high-fat diet-induced obese mouse models. The study will also examine its impact on adipose tissue remodeling, energy expenditure, and overall metabolic function. Our long-term objective is to explore the potential of nobiletin as a functional bioactive compound in nutritional interventions or nutraceutical products, particularly targeting metabolic disorders such as obesity and type 2 diabetes.